# Description of a New Species and New Records of Naucoridae (Hemiptera: Heteroptera: Nepomorpha) from Eastern Brazil [note 1]

**DOI:** 10.3390/insects15060383

**Published:** 2024-05-23

**Authors:** Rafael Jordão, Juliana da Costa Santos, Julianna Freires Barbosa, Felipe Ferraz Figueiredo Moreira

**Affiliations:** 1Laboratório de Entomologia, Instituto Oswaldo Cruz, Fundação Oswaldo Cruz, Rio de Janeiro 21040-360, Brazil; ppmeiameiameia@gmail.com; 2Laboratório de Entomologia, Departamento de Zoologia, Instituto de Biologia, Universidade Federal do Rio de Janeiro, Rio de Janeiro 21941-902, Brazil; julianna.freires@gmail.com

**Keywords:** aquatic bugs, aquatic insects, Atlantic Forest, Caatinga, taxonomy

## Abstract

**Simple Summary:**

True bugs (Hemiptera: Heteroptera) of the infraorder Nepomorpha are known as aquatic bugs because most of them spend almost their entire life cycle submerged. Naucoridae are the third most diverse family of aquatic bugs, including eight subfamilies, 43 genera, and over 400 species. They are commonly known as saucer bugs, and occur in standing and flowing freshwater courses globally, except for Antarctica and the Pacific oceanic islands. The family is represented in Brazil by 68 species, most of which are endemic. Although the diversity of the family has been the target of several recent studies, large areas of the country are still unexplored and there are several species unknown to science. Aiming to fill these knowledge gaps, a series of expeditions were carried out in seven states of eastern Brazil between 2018 and 2023. In addition, the examination of specimens previously deposited in a national entomological collection resulted in the discovery of a new species and new geographic distribution records for 11 other species and two subspecies of the family.

**Abstract:**

The aquatic bug family Naucoridae (Hemiptera: Heteroptera: Nepomorpha) is currently represented in Brazil by 68 species. Although the diversity of the family has been the target of several recent studies, large areas of the country are still unexplored and several species that have been deposited in entomological collections are waiting for a formal description. Aiming to fill these knowledge gaps, a series of expeditions were carried out in six states of eastern Brazil between 2018 and 2023: Alagoas, Bahia, Ceará, Espírito Santo, Pernambuco, and Sergipe. The fieldwork targeted protected areas, but surrounding regions were also explored. The material obtained, in addition to specimens previously deposited in a national entomological collection, revealed the existence of *Australambrysus margaritifer* Jordão, Santos and Moreira, a new species herein described, and new records for other 11 species and two subspecies belonging to the genera *Carvalhoiella* De Carlo, 1963, *Limnocoris* Stål, 1876, *Maculambrysus* Reynoso-Velasco and Sites 2021, and *Pelocoris* Stål, 1876.

## 1. Introduction

Naucoridae (Hemiptera: Heteroptera: Nepomorpha), or saucer bugs, are commonly found in lotic and lentic environments globally, except for Antarctica and the Pacific oceanic islands [1,2]. They are the third most diverse family of aquatic bugs, including eight subfamilies, 43 genera, and over 400 species [3]. Saucer bugs can be easily recognized by the dorsoventrally flattened body, hemelytral membrane without venation, well-developed anterior femora, and raptorial anterior tibiae. They are top predators in the places where they occur and influence the trophic flow of their microhabitats [4,5,6].

The following five subfamilies and ten genera are currently found in Brazil: Ambrysinae (*Australambrysus* Reynoso-Velasco and Sites, 2021; *Carvalhoiella* De Carlo, 1963; *Maculambrysus* Reynoso-Velasco and Sites, 2021; *Melloiella* De Carlo, 1935; *Pelocoris* Stål, 1876; *Picrops* La Rivers, 1952), Cryphocricinae (*Cryphocricos* Signoret, 1850), Ilyocorinae (*Placomerus* La Rivers, 1956), Laccocorinae (*Ctenipocoris* Montandon, 1897), and Limnocorinae (*Limnocoris* Stål, 1860). Limnocorinae is the richest family in the country, with 33 recorded species, followed by Ambrysinae, with 25 [7].

Despite their diversity and wide distribution in national territory, most published species records (57%) are concentrated in the southeastern (36%) and northern regions (21%), while large areas of the country remain unexplored [8]. Here, we describe a new species of *Australambrysus* from the states of Alagoas, Bahia, and Espírito Santo. Furthermore, we present new records for 11 other species and two subspecies based on material collected in seven states in the eastern portion of the country.

## 2. Materials and Methods

Most of the material studied was collected in a series of expeditions carried out in seven states of eastern Brazil between 2018 and 2023: Alagoas (AL), Bahia (BA), Ceará (CE), Espírito Santo (ES), Pernambuco (PE), and Sergipe (SE). The fieldwork targeted the following protected areas, as well as adjacent regions: Estação Ecológica de Murici (EEM, AL), Reserva Biológica de Pedra Talhada (RBPT, AL/PE), Reserva Extrativista Marinha da Lagoa do Jequiá (RESEX, AL), Parque Nacional da Chapada Diamantina (PNCD, BA), Estação Ecológica de Aiuaba (EEA, CE), Monumento Natural dos Pontões Capixabas (MNPC, ES), Reserva Biológica de Comboios (RBC, ES), Reserva Biológica de Sooretama (RBS, ES), Reserva Biológica do Córrego do Veado (RBCV, ES), Parque Nacional do Catimbau (PNCA, PE), and Parque Nacional da Serra de Itabaiana (PNSI, SE).

These expeditions have been part of the projects “Diversity and distribution of aquatic bugs (Insecta: Heteroptera: Gerromorpha & Nepomorpha) from Alagoas e Sergipe, northeastern Brazil” (Conselho Nacional de Desenvolvimento Científico e Tecnológico (CNPq), process #429936/2016-8; Sistema de Autorização e Informação em Biodiversidade (SISBIO), permits #60275-1 and #60275-2); “Diversity and conservation of Hemiptera (Insecta) from the Caatinga” (CNPq, process #421.413/2017-4; SISBIO, permits #62159-1, #62159-2, #62159-3, #62159-4, and #62159-5); and “Diversity and distribution of aquatic bugs (Hemiptera: Heteroptera: Nepomorpha) from the state of Espírito Santo, southeastern Brazil” (Universidade Federal Rural do Rio de Janeiro, masters’ dissertation project; SISBIO, permit 82288-1).

The geographic coordinates of the collecting sites were obtained with a GPS receiver. Individuals were collected by active search, with the aid of aquatic D-nets and sieves, in lentic and lotic water bodies such as swamps, ponds, pools, streams, and rivers. Specimens have been preserved in ethanol at 96% or higher concentrations, and deposited in the following institutions: Coleção Entomológica do Instituto Oswaldo Cruz, Fundação Oswaldo Cruz, Rio de Janeiro, Brazil (CEIOC); and Coleção Zoológica do Maranhão, Universidade Estadual do Maranhão, Caxias, Brazil (CZMA). Additional material previously deposited in the Coleção Entomológica Prof. José Alfredo Pinheiro Dutra, Universidade Federal do Rio de Janeiro, Rio de Janeiro, Brazil (DZRJ), and in the Coleção Zoológica Norte Capixaba, Universidade Federal do Espírito Santo, São Mateus, Brazil (CZNC), now in the Museu de Entomologia, Universidade Federal de Viçosa, Viçosa, Brazil (UFVB), was also examined. Genera were identified using keys provided by Moreiral et al. [9], and the taxonomy was updated according to the latest publications, for example, [3,10]. Species were identified using the literature specialized in each genus.

Digital photographs of the specimens were obtained using the Leica M205 C stereomicroscope under different focal distances, stacked and compacted into single images using the Leica Application Suite V4.6. Image enhancements were made using Adobe Photoshop CC 2015. All measurements are given in millimeters. In the geographic distribution section of represented species, only the first known reference is cited for each country or territory. In contrast, all known references are cited for the distribution in Brazil. Abbreviations of Brazilian federal units follow the official standard [11]. Distribution maps have been produced using QGIS v. 3.16.3 [12].

## 3. Results

### 3.1. New Species

*Australambrysus margaritifer* Jordão, Santos and Moreira, sp. nov. (Figure 1, Figure 2, Figure 3, Figure 4 and Figure 5)

urn:lsid:zoobank.org:act:94FA5B3A-7B11-4750-B4C3-6F8C5C7BF935

Type material (all specimens macropterous). Holotype. BRAZIL—Alagoas • Viçosa, Baixa Funda farm, stream; −09.3243, −36.2830; 21.V.2019; H.D.D. Rodrigues, W. Sousa, F.F.F. Moreira & J.M.S. Rodrigues leg.; ♂, CZMA. Paratypes. BRAZIL—Alagoas • same data as holotype; 1 ♂, 2 ♀, CEIOC 83191.—Bahia • Itaeté, PNCD, Timbó River; alt. 428 m; −12.6036, −41.5250; 07.V.2021; J.M.S. Rodrigues & J.F. Barbosa leg.; 3 ♂, 1 ♀, CEIOC 83192 • Mucugê, PNCD, Una River; −13.2921, −41.2524; 11.V.2021; J.F. Barbosa leg.; 1 ♂, CEIOC 83193.—Espírito Santo • Domingos Martins, waterfall; −20.3626, −40.6582; 28.III.2010; F.F. Salles leg.; 1 ♂, UFVB • Nova Venécia, Santa Rita do Pip Nuck; −18.71, −40.54; 02.VI.2014; F.F. Salles leg.; 1 ♂, 2 ♀, UFVB • São Gabriel da Palha, Parque da Ilha; −19.0475, −40.5930; 15.IX.2013; F.F. Salles leg.; 1 ♀, UFVB.

Macropterous male. Body length 9.13; maximum body width 5.29. General aspect oblong, wider across embolia. Head and pronotum light-brown; pronotum with dark-brown marks, roughly punctate. Hemelytra dark-brown, with yellowish spots. Anterolateral margins of abdominal laterotergites dark-brown. Venter yellowish-brown; abdomen with yellowish pubescence.

Head. Length 1.47; maximum width 2.03. Dorsally light-brown, with dark median stripe extending to posterior region; roughly punctate; anterior margin convex, extending through 10.21% of head length; posterior margin strongly convex, extended posteriorly 37.41% of head length. Eyes converging anteriorly. Synthilipsis 1.23. Labrum rounded anteriorly; width about three times length. Proximal three labial articles light-brown; distal article dark-brown. Antennal proportions 1:2:4:1.3; total length 0.48, not reaching lateral eye margin; long setae on two distal articles.

Thorax. Pronotum roughly punctate, brown, with dark-brown marks; posterior region light-brown, delimited anteriorly by a dark-brown stripe; anterior margin deeply concave, surrounding convex posterior margin of head between eyes; posterolateral margins slightly convex; 2.9 times wider than long; median length 1.46; maximum width on posterolateral region 4.37. Prothorax ventrally pruinose, except pubescent sides. Propleura yellowish-brown; posterior margin with golden setae; apices uniting medially (Figure 5). Probasisternum carinated. Prosternelum covered by apices of propleura. Scutellum strongly punctate, triangular, isosceles; 1.93 times wider than long; length 1.47; width 2.84; base dark-brown; apex light-brown. Hemelytra dark-brown, densely punctate; length from proximal margin of embolium to apex of membrane 6.71; corium and clavus dark-brown, with yellow punctation; corium 2.37 times longer than wide, length 4.79, width 2.02; claval commissure light-brown, length 0.99. Embolium yellowish-brown proximally; dark-brown distally; length 2.82; maximum width 0.72; lateral margin convex. Membrane dark-brown. Hindwing reaching posterior margin of abdominal tergum VI. Mesobasisternum with small tumescence, grooved; triangular throughout mesosternelum. Metasternelum (=metaxyphus) subtriangular, with small carina anteriorly (Figure 2). Legs yellowish-brown. Anterior coxa with group of spiniform setae anteriorly. Posterior margin of anterior femur with golden setae proximally; distally, with golden setae interspersed by glabrous portion; anterior margin with cluster of short golden setae. Anterior tibia and tarsus with occlusal face concave; tarsus immobile, monoarticulated, with single small claw. Middle and posterior coxae partially inserted between thoracic ventrites. Middle and posterior femora with short light-brown spines evenly distributed along anterior margin. Middle tibia with dorsolateral and ventrolateral rows of light-brown spurs, and mesolateral rows of spurs interspersed with spiniform combs. Middle and hind tibia with semicircular apical rows of spurs. Middle and hind tarsi with profuse long golden setae throughout posterior face. Middle and hind pretarsi each with pair of slender claws slightly curved at apex.

Length. Anterior leg: femur 2.28; tibia 1.99; tarsus 0.38. Middle leg: femur 2.26; tibia 2.79; tarsomeres I–III 0.10, 0.48, 0.36. Posterior leg: femur 2.66, tibia 3.12, tarsomeres I–III 0.20, 0.53, 0.50.

Abdomen. Lateral margins smooth. Posterolateral margins of segments III–VII visible in dorsal view; III–VI with anterior third dark-brown, light-brown posteriorly. Posterolateral region of segment II, in ventral view, at right angle; acute at segments III–VII, each segment with a conspicuous spiniform process. Acessory process of tergum VI absent. Median lobes of tergum VIII (pseudoparameres) asymmetrical; left lobe slightly thinner, with posterolateral margin rounded, posteromesal margin concave, and posterior margin convex (Figure 4B). Ventrites yellowish-brown, pubescent, with short golden setae (Figure 2A); lateral margins with narrow glabrous stripe. Spiracles at laterosternites II–VI surrounded by glabrous oval areas. Aedeagus long, slender; apex acute, abruptly narrowed (Figure 4D). Parameres symmetrical, slender; apical lobes longer than wide; mesal margin concave, with long setae (Figure 4A). Proctiger approximately three times longer than wide. Pygophore with anterior margin medially obtuse, covered with long setae, with thicker setae on posterior margin (Figure 4A,C).

Macropterous female. Body length 9.57; maximum body width 5.53. Left abdominal laterosternite VI without spine on posterior margin; VII with posterior margin tapered, surpassing posterior end of subgenital plate. Subgenital plate symmetrical, rhomboid (Figure 3); width 0.89 times length; length at midline 2.95; maximum width 2.63; posterolateral margins rounded (Figure 1B).

Comments. This new species is part of the *Australambrysus plax* complex, based on the propleura strongly adhered to the probasisternum and prosternelum (Figure 5) [3]. Within the complex, it is more similar to *A. aguaro* Sites, 2023, based on the margins of the anterolateral region of the pronotum, and the shape of the male abdominal tergum VIII and aedeagus. However, males of the two species can be distinguished by the following combination of characteristics: pronotum width:length ratio larger in *A. aguaro* (3.4 vs. 2.9 in *A. margaritifer*); smaller body size in *A. aguaro* (length 7.12, maximum width 4.08 vs. 9.13 and 5.29 in *A. margaritifer*); apex of pygophore tapered in *A. aguaro* (vs. obtuse in *A. margaritifer*); and parameres robust and wide in *A. aguaro* (vs. long and slender in *A. margaritifer*). Females of the new species do not bear spine on left abdominal laterosternite VI and have posteriorly acute laterosternites VII surpassing the posterior end of the subgenital plate (Figure 3B). In females of *A. aguaro*, the spiniform process is present on the left abdominal laterosternite VI.

Etymology. The specific epithet is a noun in apposition and refers to the white spots on the hemelytra of this species (*margarita* lat. = pearl; *fer* lat. = to bear). Additionally, the name is also in honor of the late mother of the first author, Margarete Jordão.

Distribution. Eastern Brazil, in the states of Alagoas, Bahia, and Espírito Santo.

### 3.2. New Records

#### 3.2.1. Ambrysinae

*Carvalhoiella beckeri* De Carlo, 1963, Rev. Soc. Entomol. Arg. 24: 11

(Figure 6A and Figure 7)

Material examined. BRAZIL—Espírito Santo • Aracruz, Monte Serrat; −19.7500, −40.3100; 13.VIII.2011; F.F. Salles leg.; 2 ♂, UFVB.

Distribution. BRAZIL: ES [present study], MG [13,14,15], MT [15].

*Maculambrysus transversus* Rodrigues, Canejo and Sites, 2024, Zootaxa 5447: 216–219

(Figure 6B, Figure 7 and Figure 8)

Material examined. BRAZIL—Alagoas • Coité do Nóia, Taquarana, waterfall; −09.6723, −36.4987; 06.VII.2018; C.F.B. Floriano, J.F. Barbosa & J.M.S. Rodrigues leg.; 1 ♂, CEIOC 83181 • Murici, EEM; −09.2202, −35.8702; 29.VII.2018; C.F.B. Floriano, J.F. Barbosa & J.M.S. Rodrigues leg.; 1 ♂, CEIOC 83156 • Murici, EEM, Vale do Socorró, unnamed stream in oil palm area; −09.2386, −35.8649; 22.V.2019; W. Sousa, F.F.F. Moreira & J.M.S. Rodrigues leg.; 1 ♂, CEIOC 83187 • Quebrangulo, RBPT, Cafuringa River; −09.2449, −36.4217; 04.VII.2018; C.F.B. Floriano, J.F. Barbosa & J.M.S. Rodrigues leg.; 2 ♂, 2 ♀, CEIOC 83172 • Viçosa, Baixa Funda Farm, stream; −09.3243, −36.2830; 21.V.2019; H.D.D. Rodrigues, W. Sousa, F.F.F. Moreira & J.M.S. Rodrigues leg.; 1 ♂, CEIOC 83188.—Bahia • Lençóis, PNCD; −12.5865, −41.3822; 22.VIII.2018; F.F.F. Moreira & J.M.S. Rodrigues leg.; 1 ♂, CEIOC 83180.—Espírito Santo • Colatina, Pancas River; −19.2986, −40.7176; 25.VI.2022; J.C. Santos, J.M.S. Rodrigues, N.O. Paiva, L. Nery & C.L. Rodrigues leg.; 1 ♀, CEIOC 83209 • Ecoporanga, river; −18.3578, −40.8824; 21.IV.2023; J.M.S. Rodrigues, N.O. Paiva, F.F.F. Moreira, I.S. Medeiros & C.L. Rodrigues leg.; 1 ♂, 2 ♀, CEIOC 83210 • Guaçuí, pond; −20.8887, −41.6058; 20.III.2022; J.C. Santos, J.M.S. Rodrigues, N.O. Paiva, & B. Clarkson leg.; 1 ♀, CEIOC 82773 • Ibitirama, Mr. Menário property, under rock; −20.4058, −41.7266; 21.IV.2008; F.F. Salles leg.; 1 ♂, UFVB • Linhares, Reserva Natural Vale, Pau Atravessado Stream, leaf litter; −19.1374, −40.0636; 17–18.VII.2011; F.F. Salles leg.; 3 ♂, UFVB • Pancas, sandy stream; −19.0989, −40.8129; 24.VI.2022; J.C. Santos, J.M.S. Rodrigues, N.O. Paiva, L. Nery & C.L. Rodrigues leg.; 5 ♂, 5 ♀, 1 nymph, CEIOC 83211 • Pedro Canário, river; −18.2239, −39.9447; 22.IV.2023; J.M.S. Rodrigues, N.O. Paiva, F.F.F. Moreira, I.S. Medeiros & C.L. Rodrigues leg.; 1 ♀, CEIOC 83212 • São Mateus, Nestor Gomes, Tapuia River, road ES-130 km 41–42; −18.7303, −40.2230; 09.II.2010; N. Ferreira-Jr. leg.; 6 ♂, 4 ♀, DZRJ • Sooretama, RBS, Quirininho Stream; −19.05, −40.14; 14–15.VII.2008; F.F. Salles leg.; 8 ♂, 2 ♀, UFVB • same, except Paraisópolis stream; 2 ♂, UFVB.—Sergipe • Areia Branca, PNSI, Vermelho Stream; −10.7387, −37.3354; 08.VII.2018; C.F.B. Floriano, J.F. Barbosa & J.M.S. Rodrigues leg.; 1 ♂, CEIOC 83183.

Distribution. BRAZIL: AL [16, present study], BA [present study], ES [present study], PA [16], RN [16], SE [present study].

*Pelocoris binotulatus binotulatus* (Stål, 1860)

*Naucoris binotulatus* Stål, 1860, Kungl. Svensk. Vetensk. Hand. 2(7): 83

(Figure 6C and Figure 7)

Material examined. BRAZIL—Espírito Santo • Aracruz, do Norte River; alt. 35 m; −19.5858, −40.2131; 28.VI.2022; J.C. Santos, J.M.S. Rodrigues, N.O. Paiva, L. Nery & C.L. Rodrigues leg.; 1 ♀, CEIOC 82776 • Sooretama, RBS, weir; alt. 503 m; −19.0354, −40.1584; 26.VI.2022; J.C. Santos, J.M.S. Rodrigues, N.O. Paiva, L. Nery & C.L. Rodrigues leg.; 1 ♀, CEIOC 82768.

Distribution. ARGENTINA [17]; BOLIVIA [18]; BRAZIL: AM [19], ES [present study], RJ [17,20,21]; COLOMBIA [19]; PARAGUAY [22]; PERU [3].

*Pelocoris binotulatus nigriculus* Berg, 1879, Hemiptera Argentina: 188

(Figure 6D, Figure 7 and Figure 8)

Material examined. BRAZIL—Alagoas • Igaci, Road BR-316, Lunga River; −09.5311, −36.5332; 06.VII.2018; J.M.S. Rodrigues, J.F. Barbosa & C.F.B. Floriano leg.; 1 adult (sex undetermined), CEIOC 83124 • Lagoa do Jequiá, RESEX; −10.0041, −36.0256; 30.IV.2018; J.M.S. Rodrigues, O.M. Magalhães & C.F.B. Floriano leg.; 1 adult (sex undetermined), CEIOC 83138 • Maravilha, Road BR-316, weir; alt. 294 m; −09.2340, −37.4512; 24.V.2019; J.M.S. Rodrigues, W. Sousa & F.F.F. Moreira leg.; 2 ♀, CEIOC 83143 • Maribondo, Flexeiras, weir; alt. 122 m; −09.5074, −36.2340; 23.V.2019; J.M.S. Rodrigues, W. Sousa & F.F.F. Moreira leg.; 2 ♀, CEIOC 83143 • Minador do Negrão, weir; alt. 253 m; −09.3679, −36.8427; 23.V.2019; J.M.S. Rodrigues, W. Sousa & F.F.F. Moreira leg.; 2 ♀, 7 adults (sex undetermined), 15 nymphs, CEIOC 83140.—Ceará • Aiuaba, EEA, Casa da Gameleira, weir; alt. 532 m; −06.6992, −40.2908; 09.IV.2019; R. Carrenho leg.; 2 adults (sex undetermined), 1 nymph, CEIOC 83123 • same, except flooded area next to road to Casa do Cajueiro; alt. 508 m; −06.6992, −40.1873; 10.IV.2019; J.M.S. Rodrigues leg.; 5 adults (sex undetermined), CEIOC 83139 • same, except Volta de Cima Ranch, ICMBio headquarters, well; alt. 435 m; −06.6019, −40.1248; 1 adult (sex undetermined), CEIOC 83136 • same, except Letreiro Weir; alt. 475 m; −06.6163, −40.1545; 13.IV.2019; 1 adult (sex undetermined), CEIOC 83132 • same, except Volta de Baixo Ranch, full weir; alt. 431 m; −06.6263, −40.1339; 14.IV.2019; 1 adult (sex undetermined), CEIOC 83134 • same, except Jatobá Ranch, weir; alt. 571 m; −06.7351, −40.2424; 03.VI.2021; 3 adults (sex undetermined), 3 nymphs, CEIOC 83148 • same, except weir; alt. 489 m; −06.6850, −40.1841; 04.VI.2021; J.M.S. Rodrigues & F.F.F. Moreira leg.; 5 adults (sex undetermined), 3 nymphs, CEIOC 83147 • same, except weir and stream; alt. 507 m; −06.6921, −40.1871; 04.VI.2021; J.M.S. Rodrigues & C.C. Gonçalves leg.; 11 adults (sex undetermined), CEIOC 83122 • same, except Volta de Cima Ranch, lake; alt. 434 m; −06.6019, −40.1246; 05.VI.2021; J.M.S. Rodrigues & F.F.F. Moreira leg.; 1 ♂, 1 ♀, 1 nymph, CEIOC 83129 • same, except weir; alt. 570 m; −06.7263, −40.2229; 4 adults (sex undetermined), CEIOC 83141 • same, except Serra Nova stream; alt. 436 m; −06.6026, −40.1248; 06.VI.2021; 1 adult (sex undetermined), CEIOC 83130 • same, except weir; alt. 584 m; −06.7384, −40.2475; 07.VI.2021; J.M.S. Rodrigues & C.C. Gonçalves leg.; 3 adults (sex undetermined), CEIOC 83159.—Espírito Santo • Colatina, weir; alt. 130 m; −19.3050, −40.7519; 25.VI.2022; J.C. Santos, J.M.S. Rodrigues, N.O. Paiva, L. Nery & C.L. Rodrigues leg.; 16 ♂, 12 ♀, CEIOC 82769 • same, except alt. 218 m; −19.3216, −40.5432; 1 ♀, CEIOC 82781 • Conceição da Barra, floodplain; alt. 21 m; −18.6341, −39.8141; 23.IV.2023; J.M.S. Rodrigues, N.O. Paiva, F.F.F. Moreira, I.S. Medeiros & C.L. Rodrigues leg.; 1 ♂, CEIOC 82778 • Ecoporanga; alt. 438 m; −18.3444, −40.8792; 21.IV.2023; J.M.S. Rodrigues, N.O. Paiva, F.F.F. Moreira, I.S. Medeiros & C.L. Rodrigues leg.; 1 ♀, CEIOC 83161 • same, except alt. 224 m; −18.2991, −40.7495; 4 ♂, 7 ♀, CEIOC 83162 • Itaguaçu, weir; alt. 146 m; −19.8084, −40.8279; 23.VI.2022; J.C. Santos, J.M.S. Rodrigues, N.O. Paiva, L. Nery & C.L. Rodrigues leg.; 1 ♂, CEIOC 82779 • Itapemirim, weir; alt. 3 m; −20.9944, −40.8978; 25.III.2022; J.C. Santos, J.M.S. Rodrigues, N.O. Paiva, & B. Clarkson leg.; 1 ♂, 1 ♀, CEIOC 82782 • Linhares, RBC, pools; alt. 3 m; −19.6767, −39.8976; 26.IV.2023; J.M.S. Rodrigues, N.O. Paiva, F.F.F. Moreira, I.S. Medeiros & C.L. Rodrigues leg.; 1 ♂, CEIOC 83163 • São Mateus, Preto River; −18.7356, −39.7969; 30.I.2007; F.F. Salles leg.; 1 ♂, UFVB.—Pernambuco • Tupanatinga, PNCA, temporary pool on trail to casa de farinha; alt. 814 m; −08.5615, −37.2339; 14.III.2019; J.M.S. Rodrigues & H.D.D. Rodrigues leg.; 1 adult (sex undetermined), CEIOC 83131 • same, except Brejo de São José Farm, near headquarters, temporary pool; alt. 699 m; −08.5370, −37.2200; 16.III.2019; 1 adult (sex undetermined), CEIOC 83127 • same, except temporary pool; alt. 700 m; −08.5060, −37.2237; 2 adults (sex undetermined), CEIOC 83128 • same, except Chapadão, temporary pool; alt. 957 m; −08.5244, −37.2394; 20.III.2019; 1 adult (sex undetermined), CEIOC 83137 • same, except Serrinha, Casa de Edivar, pothole with temporary water; alt. 921 m; 21.III.2019; −08.5189, −37.2335; 1 ♂, CEIOC 83144 • same, except 3 ♂, 1 ♀, 3 nymphs, CEIOC 83149 • Buíque, PNCA, Brejo de São José Farm, pond, goat weir; alt. 673 m; −08.5247, −37.1968; 16.III.2021; J.M.S. Rodrigues & R. Jordão leg.; 2 adults (sex undetermined), CEIOC 83131.—Sergipe • Areia Branca, PNSI, Negro Stream; −10.7475, −37.3402; 08.VII.2018; C.F.B. Floriano, J.F. Barbosa & J.M.S. Rodrigues leg.; 3 adults (sex undetermined), 2 nymphs, CEIOC 83125 • Canindé de São Francisco, Curitiba Village, weir full of *Pistia*; alt. 277 m; −09.7162, −37.9441; 26.VI.2019; W. Sousa, F.F.F. Moreira & J.M.S. Rodrigues leg.; 1 adult (sex undetermined), 2 nymphs, CEIOC 83145 • same, except weir; alt. 301 m; −09.8549, −37.9391; 1 ♂, 1 nymph, CEIOC 83142 • same, tributary to Curitiba River; alt. 190 m; −09.6443, −37.9185; 1 ♀, CEIOC 83160 • Lagarto, Road SE-170, weir; −10.9132, −37.6714; 09.VII.2018; C.F.B. Floriano, J.F. Barbosa & J.M.S. Rodrigues leg.; 16 adults (sex undetermined), CEIOC 83126 • Poço Redondo, Fernando Farm, weir; alt. 275 m; −09.9015, −37.5862; 26.V.2019; W. Sousa, F.F.F. Moreira & J.M.S. Rodrigues leg.; 1 adult (sex undetermined), 8 nymphs, CEIOC 83151 • same, except Jacaré River; −09.8078, −37.6859; 1 adult (sex undetermined), CEIOC 83120 • same, except 20 adults (sex undetermined), CEIOC 83133 • Porto da Folha, weir; alt. 278 m; −09.9713, −37.5907; 27.V.2019; W. Sousa, F.F.F. Moreira & J.M.S. Rodrigues leg.; 1 ♂, 1 ♀, CEIOC 83150.

Distribution. ARGENTINA [23]; BOLIVIA [24]; BRAZIL: AL [present study], CE [present study], ES [present study], MA [25], MG [26], PE [present study], SE [present study].

*Pelocoris magister* Montandon, 1898, Bull. Soc. Sci. Bucarest-Roumanie 7: 289–290

(Figure 6E, Figure 7 and Figure 8)

Material examined. BRAZIL—Alagoas • Quebrangulo, RBPT, Juliana Ranch, river; −09.2340, −36.4489; 05.VII.2018; C.F.B. Floriano, J.F. Barbosa & J.M.S. Rodrigues leg.; 1 ♀, CEIOC 83195.—Espírito Santo • Guaçuí, weir; alt. 583 m; −20.8887, −41.6958; 20.III.2022; J.C. Santos, J.M.S. Rodrigues, N.O. Paiva, & B. Clarkson leg.; 1 ♂, CEIOC 82788 • São Mateus, flooded area; −18.7714, −39.8153; 10.XI.2007; F.F. Salles leg.; 1 ♀, 4 nymphs, UFVB.

Distribution. ARGENTINA [27]; BRAZIL: AL [present study], BA [25], ES [17], MA [25], MG [28,29,30,31], PA [32], RJ [17,33,34], RS [35]; PERU [3].

*Pelocoris politus* Montandon, 1895, Boll. Mus. Zool. Anat. Comp. R. Univ. Torino 10(210): 8–9

(Figure 6F and Figure 8)

Material examined. BRAZIL—Alagoas • Piaçabuçu, Carrancas Restaurant, São Francisco River; −10.3978, −36.4462; 27.V.2019; W. Sousa, F.F.F. Moreira & J.M.S. Rodrigues leg.; 1 ♂, CEIOC 83121 • União dos Palmares, Mundaú River; −9.1342, −36.0809; 21.V.2019; W. Sousa, F.F.F. Moreira & J.M.S. Rodrigues leg.; 1 ♀, 1 nymph, CEIOC 83146.

Distribution. ARGENTINA [36]; BRAZIL: AL [present study], AM [37], MA [25], MG [30], MT [38], PA [37]; COLOMBIA [19]; FRENCH GUIANA [10]; PARAGUAY [39].

*Pelocoris subflavus* Montandon, 1898, Bull. Soc. Sci. Bucarest-Roumanie 7: 288–289

(Figure 6G and Figure 7)

Material examined. BRAZIL—Espírito Santo • Colatina, pond; alt. 130 m; −19.3050, −40.7519; 25.VI.2022; J.C. Santos, J.M.S. Rodrigues, N.O. Paiva, L. Nery & C.L. Rodrigues leg.; 2 ♂, 1 ♀, 1 nymph, CEIOC 82783 • Conceição da Barra, floodplain; alt. 21 m; −18.6341, −39.8141; 23.IV.2023; J.M.S. Rodrigues, N.O. Paiva, F.F.F. Moreira, I.S. Medeiros & C.L. Rodrigues leg.; 1 ♂, 1 ♀, CEIOC 83164 • Ecoporanga, pond; alt. 224 m; −18.2991, −40.7495; 21.IV.2023; J.M.S. Rodrigues, N.O. Paiva, F.F.F. Moreira, I.S. Medeiros & C.L. Rodrigues leg.; 2 ♀, CEIOC 83165 • same, except alt. 438 m; −18.3444, −40.8792; 21.IV.2023; 1 ♂, 2 ♀, CEIOC 83166• Guaçuí, pond; alt. 583 m; −20.8887, −41.6958; 20.III.2022; J.C. Santos, J.M.S. Rodrigues, N.O. Paiva, & B. Clarkson leg.; 1 ♂, CEIOC 82777 • Itaguaçu, pond; alt. 146 m; −19.8084, −40.8279; 23.VI.2022; J.C. Santos, J.M.S. Rodrigues, N.O. Paiva, L. Nery & C.L. Rodrigues leg.; 3 ♂, 3 ♀, CEIOC 82767 • Pancas, MNPC, pond; alt. 104 m; −19.2358, −40.7677; 24.VI.2022; J.C. Santos, J.M.S. Rodrigues, N.O. Paiva, L. Nery & C.L. Rodrigues leg.; 1 ♂, 2 ♀, CEIOC 82780 • Santa Teresa, pond; alt. 747 m; −19.8200, −40.7799; 23.VI.2022; J.C. Santos, J.M.S. Rodrigues, N.O. Paiva, L. Nery & C.L. Rodrigues leg.; 1 ♂, 2 ♀, CEIOC 82770.

Distribution. ARGENTINA [27]; BRAZIL: BA [25], ES [present study], MA [25], MG [26,28,29,31,40], MS [41], RJ [34], RS [17]; URUGUAY [42].

#### 3.2.2. Limnocorinae

*Limnocoris acutalis* La Rivers, 1974, Biol. Soc. Nevada Occ. Pap. 38: 6–7

(Figure 6H and Figure 7)

Material examined. BRAZIL—Espírito Santo • Alegre, Fumaça Waterfall, marginal vegetation; −20.6300, −41.6050; 27.V.2014; F.F. Salles leg.; 4 ♂, 1 nymph, UFVB • Conceição do Castelo, Vargas Waterfall; −20.3420, −41.2340; 31.I.2001; F.F. Salles leg.; 1 ♂, UFVB • Dores do Rio Preto;-20.6, -41.8; 25.III.2011; F.F. Salles leg.; 1 ♂, 1♀, UFVB • Pedro Canário, Assentamento Castro Alves; −18.2400, −40.0500; 29.III.2011; F.F. Salles leg.; 1 ♂, UFVB.

Distribution. BRAZIL [43]: ES [present study], MG [3], RJ [44].

*Limnocoris brasiliensis* De Carlo, 1941, Rev. Soc. Entomol. Arg. 11: 37–38

(Figure 6I and Figure 7)

Material examined. BRAZIL—Espírito Santo • Ecoporanga, Bonita Waterfall; −18.3280, −40.7940; 30.VI.2015; F.F. Salles leg.; 1 ♀, UFVB.

Distribution. BRAZIL [45,46]: ES [present study], MG [5,47,48], RJ [5,34,47,49,50,51], SP [5,52,53].

*Limnocoris insignis* Stål, 1860, Kungl. Svensk. Vetensk. Hand. 2(7): 83

(Figure 6J and Figure 7)

Material examined. BRAZIL—Espírito Santo • Santa Teresa, Nova Lombardia, Capitel de Santo Antônio, Escavado Stream; −19.8755, −40.5298; 24–25.X.2008; F.F. Salles leg.; 2 ♂, UFVB.

Distribution. BRAZIL [5,54]: ES [present study], MG [3,5], PR [5], RJ [5,20,21,46,55,56,57], RS [5], SC [5], SP [5,43].

*Limnocoris pusillus* Montandon, 1897, Boll. Mus. Zool. Anat. Comp. R. Univers. Torino 12(297): 7–8

(Figure 6K, Figure 7 and Figure 8)

Material examined. BRAZIL—Alagoas • Viçosa, Anel Waterfall; alt. 333 m; −09.3175, −36.2944; 21.V.2019; H.D.D. Rodrigues, W. Sousa, F.F.F. Moreira & J.M.S. Rodrigues leg.; 1 adult (sex undetermined), CEIOC 83113.—Espírito Santo • Afonso Cláudio, Santa Luzia Waterfall; −20.1600, −41.1400; 26.III.2011; F.F. Salles leg.; 1 ♂, UFVB • Aracruz, César Farm; alt. 44 m; −19.7287, −40.2487; 14.X.2010; F.F. Salles leg.; 1 ♀, UFVB • Conceição da Barra, stream; −18.4327, −40.0152; 22.IV.2023; J.M.S. Rodrigues, N.O. Paiva, F.F.F. Moreira, I.S. Medeiros & C.L. Rodrigues leg.; 1 ♂, 4 ♀, CEIOC 83168 • same, except −18.5800, −39.7300; 01.IV.2008; F.F. Salles leg.; 5 ♂, 3 ♀, 3 nymphs, UFVB • Linhares, Reserva Natural Vale, Pau Atravessado Stream; −19.1372, −40.0675; 14.VIII.2011; F.F. Salles leg.; 4 ♂, 3 ♀, 9 nymphs, UFVB• Muniz Freire, Norte River; −20.3799, −41.4051; 23.III.2022; J.C. Santos, J.M.S. Rodrigues, N.O. Paiva, & B. Clarkson leg.; 1 ♀, CEIOC 83167 • Pancas, Giles Waterfall; −19.13, −40.79; 25.VI.2012; F.F. Salles leg.; 2 ♂, 1 ♀, UFVB • same, except MNPC, Dona Laura Waterfall; −19.1233, −40.8144; 24.VI.2022; J.C. Santos, J.M.S. Rodrigues, N.O. Paiva, L. Nery & C.L. Rodrigues leg.; 1 ♀, CEIOC 82771 • Pinheiros, RBCV; alt. 66 m; −18.3540, −40.1358; 18.I.2010; F.F. Salles leg.; 3 ♂, 3 ♀, UFVB • same, except 24.II.2010; 2♀, 2 nymphs, UFVB • same, except 21.IV.2010; 1 ♂, 1 ♀, UFVB • same, except stream; alt. 82 m; −18.3678, −40.1398; 24.IV.2023; J.M.S. Rodrigues, N.O. Paiva, F.F.F. Moreira, I.S. Medeiros & C.L. Rodrigues leg.; 9 ♂, 2 ♀, 1 nymph, CEIOC 83169 • Sooretama, RBS, Rodrigues Stream; alt. 44 m; −19.0268, −40.2275; 08.II.2011; F.F. Salles leg.; 2 ♂, 1 ♀, UFVB.

Distribution. ARGENTINA [5]; BOLIVIA [46]; BRAZIL: AL [58; present study], AM [58], ES [present study], GO [58], MG [5,26,28,31,58,59,60], MT [58,61,62], RJ [5,34,54,58], RS [5,58,63], SC [5,58], TO [58]; COLOMBIA [19]; FRENCH GUIANA [64]; PARAGUAY [58]; PERU [58]; VENEZUELA [58].

*Limnocoris volxemi* (Lethierry, 1877)

*Borbocoris volxemi* Lethierry, 1877, Ann. Soc. Entomol. Belgique 20: 41

(Figure 6L and Figure 8)

Material examined. BRAZIL—Bahia • Mucugê, PNCD, Véu da Noiva Waterfall; alt. 776 m; −13.2881, −41.2683; 06.V.2021; J.M.S. Rodrigues & J.F. Barbosa leg.; 2 adults (sex undetermined), CEIOC 83119 • Palmeiras, PNCD, Preto River; alt. 820 m; −12.6036, −41.5250; 05.V.2021; J.M.S. Rodrigues & J.F. Barbosa leg.; 7 adults (sex undetermined), CEIOC 83112.

Distribution. BRASIL [5,17,65,66]: BA [present study], MG [3,5,28,30,31,46,48,59,60,67,68], MT [69], PR [5], SC [5].

## 4. Discussion

Based on the collecting effort in eastern Brazil, we discovered and described a new species of the recently proposed genus *Australambrysus*, i.e., *A. margaritifer* Jordão, Santos and Moreira. We also expanded the known distributions of previously described species and provided several first state records. Our findings indicate that fieldwork in the states of northeastern Brazil that have not yet been properly explored might yield relevant new biodiversity data, decreasing the Linnean and Wallacean shortfalls. Furthermore, our results fill distribution gaps in the Atlantic Forest of Espírito Santo, which had been very poorly explored compared with the other three states in southeastern Brazil—Minas Gerais, Rio de Janeiro, and São Paulo. Finally, much of our examined material was collected in protected areas in northeastern and southeastern Brazil, reinforcing the importance of their effective management and protection to conserve the aquatic insect fauna.

## Figures and Tables

**Figure 1 insects-15-00383-f001:**
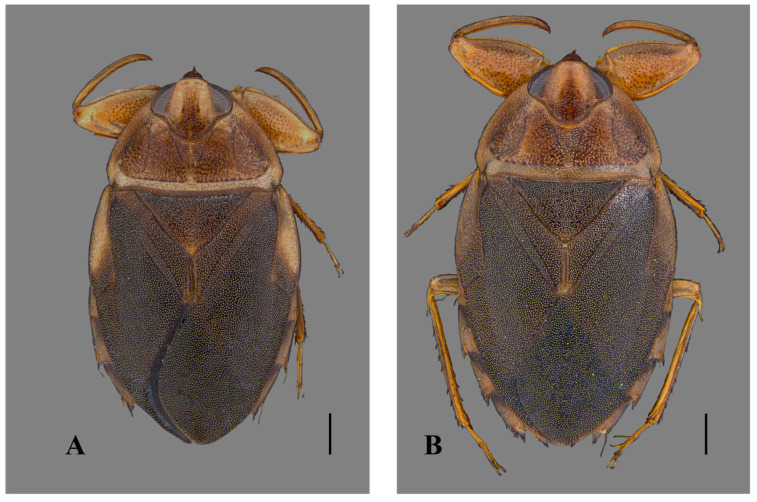
*Australambrysus margaritifer* Jordão, Santos and Moreira, sp. nov., dorsal habitus. (**A**) Male; (**B**) female. Scale bars: 1 mm.

**Figure 2 insects-15-00383-f002:**
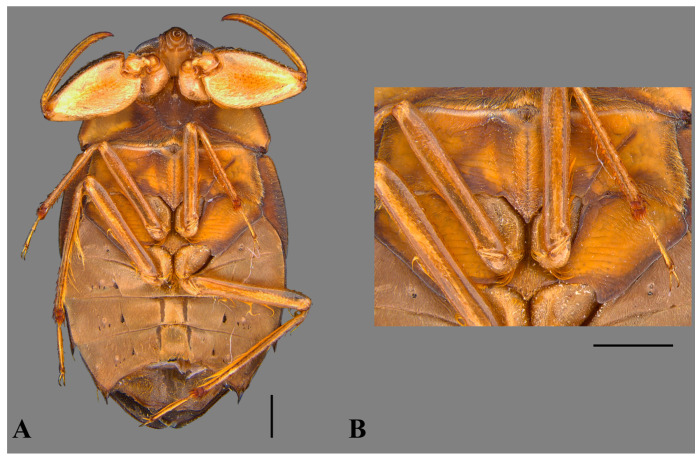
*Australambrysus margaritifer* Jordão, Santos and Moreira, sp. nov., male, ventral view. (**A**) Habitus, terminalia removed; (**B**) detail of central portion of thorax. Scale bars: 1 mm.

**Figure 3 insects-15-00383-f003:**
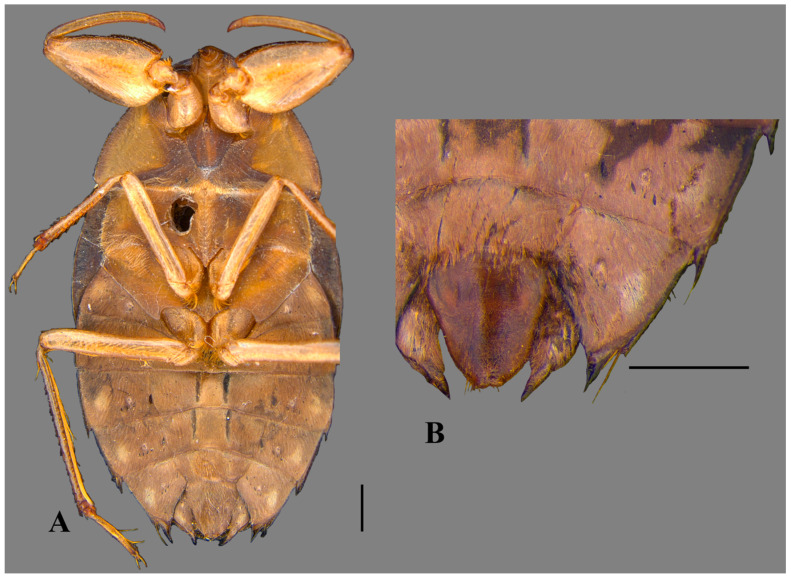
*Australambrysus margaritifer* Jordão, Santos and Moreira, sp. nov., female, ventral view. (**A**) Habitus; (**B**) apex of abdomen. Scale bars: 1 mm.

**Figure 4 insects-15-00383-f004:**
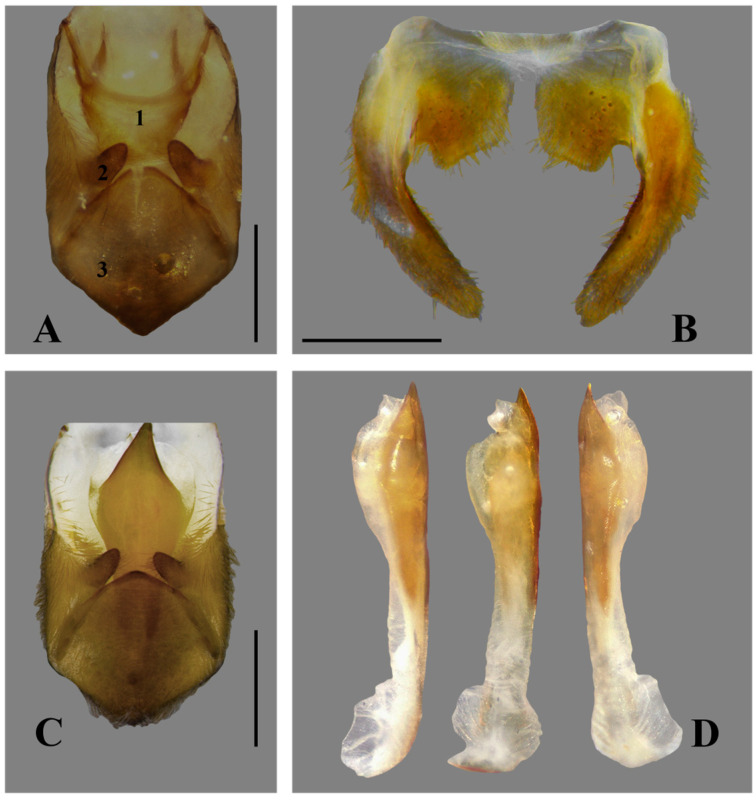
*Australambrysus margaritifer* Jordão, Santos and Moreira, sp. nov., male, terminalia. (**A**) Proctiger (1), parameres (2), and pygophore (3); (**B**) pseudoparameres (tergum VIII); (**C**) phallosome, parameres, and pygophore (proctiger removed); (**D**) phallosome. Scale bars: 1 mm.

**Figure 5 insects-15-00383-f005:**
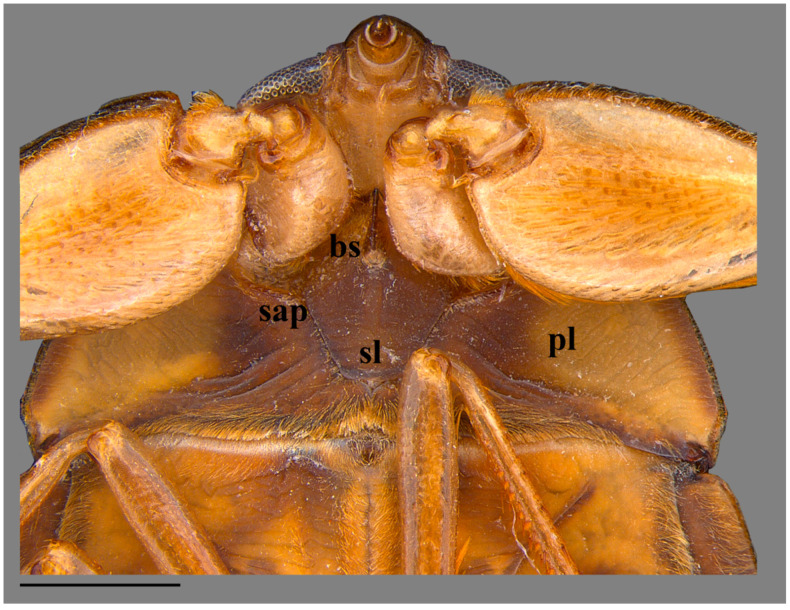
*Australambrysus margaritifer* Jordão, Santos and Moreira, sp. nov., head and anterior portion of thorax, ventral view [bs basisternum, pl pleura, sap sternal apophyseal pit, sl sternelum]. Scale bar: 1 mm.

**Figure 6 insects-15-00383-f006:**
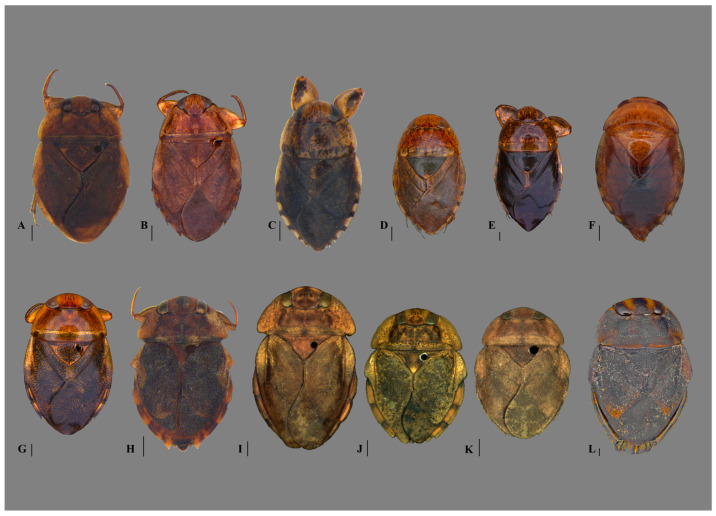
Naucoridae from eastern Brazil, dorsal habitus. (**A**) *Carvalhoiella beckeri*; (**B**) *Maculambrysus transversus*; (**C**) *Pelocoris binotulatus binotulatus*; (**D**) *P. b. nigriculus*; (**E**) *P. magister*; (**F**) *P. politus*; (**G**) *P. subflavus*; (**H**) *Limnocoris acutalis*; (**I**) *L. brasiliensis*; (**J**) *L. insignis*; (**K**) *L. pusillus*; (**L**) *L. volxemi*. Scale bars: 1 mm.

**Figure 7 insects-15-00383-f007:**
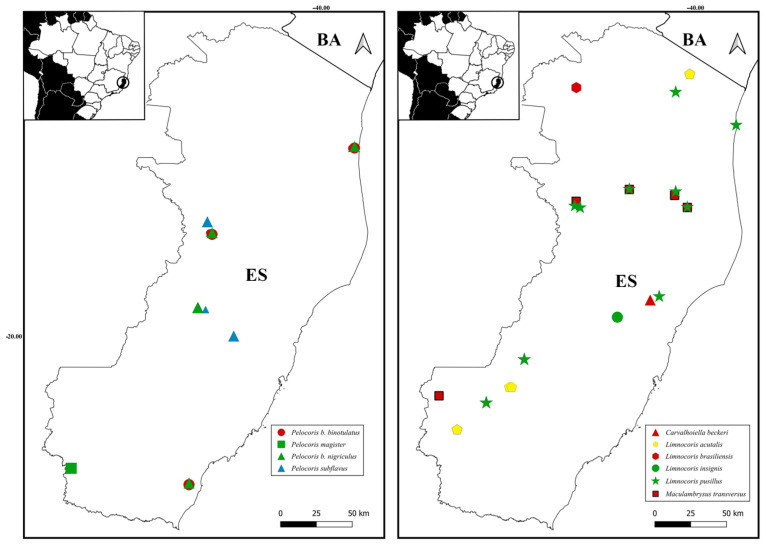
Geographic distribution of the new records from the state of Espírito Santo (ES).

**Figure 8 insects-15-00383-f008:**
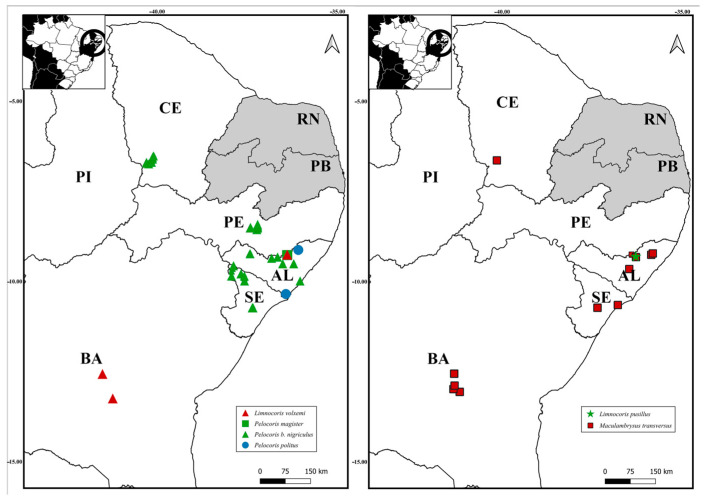
Geographic distribution of the new records from the states of Alagoas (AL), Bahia (BA), Ceará (CE), Pernambuco (PE), and Sergipe (SE).

## Data Availability

The data presented in this study are available in the article.

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
