# Peer review of "Description of a New Species and New Records of Naucoridae (Hemiptera: Heteroptera: Nepomorpha) from Eastern Brazilâ€"

_insects, 2024, doi:10.3390/insects15060383_

Round 1
Reviewer 1 Report
Comments and Suggestions for Authors
I congratulate the authors on their very good and correct work. I have no objections to the description of the new species. The list of new places of occurrence for the described species is impressive. Actually, it complements the state of knowledge about the distribution of representatives of the discussed family in the territory of Brazil.
I only have a few language suggestions, which are included in the attached PDF file. The only thing missing in this work is information about ecology, hunting methods, possible victims, etc. Any observations of this type that the authors could have made during so many years of research. Perhaps you have such observations?
After considering these most important linguistic corrections, I recommend that the publication be accepted for publication.

Comments on the Quality of English LanguageJust a few minor corrections to be made
Author Response
All linguistic corrections suggested by the reviewer have been followed.
"The only thing missing in this work is information about ecology (...). Perhaps you have such observations?"
Unfortunately, our objective during the fieldwork was purely taxonomic, so we do not have detailed information about ecology or other aspects.
Reviewer 2 Report
Comments and Suggestions for Authors
This manuscript brings important information on nepomorphan fauna in Brazil, including new records for several Brazilian states and also Brazil as a whole. Most importantly, a new species, Australambrysus margaritifer, has been described. The methods, deployed by the authors, include standard entomological sampling during several expeditions and definitely could result in the samples referred by the authors. The description of the new species is standard, includes all neccesarry information and is totally comparable to other descriptions of the new species. The figures - pictures of important determination details of the new species and maps of the sampling sites - are of good quality and fittingly complement the description and the whole manuscript. I have already suggested adding the only missing information (ecology of the new species, at least habitat description). The conclusions stated by authors are consistent with the results, the results contribute to the knowledge of the Brazilian aquatic entomofauna (and aquatic entomofauna of the neotropics in general). References cited in the manuscript cover sufficiently scientific background of the manuscript.
Author Response
"I only suggest adding information on habitat or fundamental ecological parameters of the site with Australambrysus margaritifer (type of water body, extent of submerged vegetation, water bottom characteristics etc.)."
Unfortunately, our objective during the fieldwork was purely taxonomic, so we do not have detailed information about ecology or other aspects.
Reviewer 3 Report
Comments and Suggestions for Authors
This a useful piece of research. Please see a few language suggestions on the attached copy of the manuscript.

Author Response
We followed all suggestions, except for "You should ad "mm" to all your measurement data."
We already mention in the Materials and Methods that "All measurements are given in millimeters." There is no need to repeat this information in the species description.